# Field-Based Evaluation of Rice Genotypes for Enhanced Growth, Yield Attributes, Yield and Grain Yield Efficiency Index in Irrigated Lowlands of the Indo-Gangetic Plains

Sandeep Gawdiya [1], Dinesh Kumar [1,*], Yashbir S. Shivay [1], Arti Bhatia [2], Shweta Mehrotra [1,*], Mandapelli Sharath Chandra [3], Anita Kumawat [1], Rajesh Kumar [1], Adam H. Price [4], Nandula Raghuram [5], Himanshu Pathak [6] and Mark A. Sutton [7]

1 Division of Agronomy, ICAR-Indian Agricultural Research Institute (IARI), New Delhi 110012, India; ysshivay@hotmail.com (Y.S.S.)
2 Division of Environment Science, ICAR-Indian Agricultural Research Institute (IARI), New Delhi 110012, India
3 ICAR-AICRP on Integrated Farming System, Professor Jayashankar Telangana Agricultural University, Rajendranagar, Hyderabad 500030, India
4 Room 1:07 Cruickshank Building, School of Biological Sciences, University of Aberdeen, Aberdeen AB24 3FX, UK
5 Centre for Sustainable Nitrogen and Nutrient Management, University School of Biotechnology, Guru Gobind Singh Indraprastha University, Dwarka, New Delhi 110078, India
6 Indian Council of Agricultural Research, New Delhi 110001, India
7 UK Centre for Ecology & Hydrology, Edinburgh Research Station, Bush Estate, Penicuik EH26 0QB, UK
* Correspondence: dineshctt@yahoo.com (D.K.); shwetamehrotragoyal@gmail.com (S.M.)

**Abstract:** Nitrogen (N) fertilizers are widely used worldwide to increase agricultural productivity. However, significant N losses contributing to air and water pollution ultimately reduce the nitrogen use efficiency (NUE) of crops. Numerous research studies have emphasized the use of a low dose of N fertilizer, but few have focused on screening N-efficient rice genotypes. This study aimed to identify and screen ten rice genotypes that are N-use-efficient under different N fertilization treatments using the surface placement of neem-oil-coated urea: $N_0$ (control), $N_{60}$ ($\frac{1}{2}$ of recommended N), and $N_{120}$ (recommended N) for two consecutive years (2020 and 2021) under a split plot design. In both growing seasons, the application of $N_{120}$ yielded the highest panicles m$^{-2}$ (PAN = 453), filled grains panicle$^{-1}$ (FGP = 133), leaf area index (LAI = 5.47), tillers m$^{-2}$ (TILL = 541), grain yield t ha$^{-1}$ (GY = 5.5) and harvest index (HI = 45.4%) by the genotype 'Nidhi', being closely followed by the genotype 'Daya'. Four genotypes ('Nidhi', 'Daya', 'PB 1728' and 'Nagina 22'), out of the ten genotypes evaluated, responded well to different fertilization treatments with N with respect to the grain yield efficiency index (GYEI $\geq$ 1). Regarding N fertilization, $N_{60}$ and $N_{120}$ recorded the highest increase in PAN (28.5%; 41.4%), FGP (29.5%; 39.3%), test weight (29.5%; 45.3%), LAI at 30 days after transplanting (DAT) (143.7%; 223.3%), and LAI at 60 DAT (61.6%; 70.1%) when compared with $N_0$. Furthermore, the application of $N_{60}$ and $N_{120}$ improved GY and HI by 47.6% and 59.4%, and 3.4% and 6.2%, respectively, over $N_0$. Nitrogen addition ($N_{60}$ and $N_{120}$) also significantly increased the chlorophyll content at 60 DAT (8.8%; 16.3%), TILL at 60 DAT (22.9%; 46.2%), TILL at harvest (28%; 41.4%), respectively, over $N_0$. Overall, our research findings clearly indicate that 'Nidhi' and 'Daya' could be efficient candidates for improved nitrogen use, grain yield and GYEI in the Indo-Gangetic plains of India.

**Keywords:** grain yield; rice genotypes; nitrogen fertilization; grain yield efficiency index

## 1. Introduction

Rice is a heavy user of nitrogen (N) fertilizers. In India, to feed the growing populations, it has been suggested that N fertilizer consumption would need an increase of approximately 24 million tons in 2030 compared with 2022; the current total N fertilizer consumption (year 2022) is around 18.86 million tons [1,2]. India's production of rice

(milled rice) increased from 53.6 million tons in the fiscal year 1980–1981 to 120 million tons in the fiscal year 2020–2021 [3]. In soil, more than 40–50% of the applied N is lost through different mechanisms, such as ammonia ($NH_3$) volatilization, denitrification to nitrous oxide ($N_2O$) and dinitrogen ($N_2$), leaching and runoff [4–6]. These losses not only reduce the yield and economic efficiency of applied N [7], but also cause grave environmental consequences [8,9].

Due to the expansion of cultivation areas, the introduction of new cultivars, and the use of chemical fertilizers, rice yield has increased during the past 50 years, keeping pace with the world's population growth [10]. Nevertheless, the N use efficiency (NUE) of applied N is still low [11–13], which not only causes climate-change-related issues, air, and water pollution, but also causes increases in the cost of production, given the waste of N as a valuable resource [11,14]. Therefore, it is important to reduce the loss of N from agricultural land [11], and there is a need for more attention to the identification and performance of N-efficient genotypes. Rice is the key staple food for the world's poorest and undernourished people living in Asia and Africa as they cannot afford—or do not have access to—nutritious foods [15]. In the next 20 years, the world population is expected to increase by about two billion, and in Asia alone, to increase by around half of the world population [16]. A report by the CGIAR System [17] notes that with the expected growth in population and income and a decline in rice acreage, global demand for rice will continue to increase from 479 million tons of milled rice in 2014 to between 536 million and 551 million tons in 2030, with little scope for the easy expansion of agricultural land or irrigation. Furthermore, rice is a semi-aquatic plant and generally grows under flooded conditions, which makes it unique [18,19]. Special difficulties in managing N arise from this preferred habitat, including significant losses of N to water.

Numerous studies were conducted before the 21st century to improve rice nitrogen use efficiency (NUE) and yield [20–22]. Their findings showed that nutrient-efficient cultivars under field conditions can help design selection regimes and identify useful traits that are important for screening N-efficient genotypes. The knowledge of the genotypes' traits that increase NUE can be combined with the best N management practices, which would help contribute to economically viable and environmentally sustainable systems globally [23].

Different levels of N input (low, medium, high) in experimental studies have shown that significant variability is present for the use, uptake, and utilization efficiency of N. Hence, these aspects are the main areas where researchers can evaluate the response of existing genotypes at various levels of N. A number of agronomic factors in crop growth cycles affect performance and overall NUE, including the optimum N rate, appropriate N source, and timing of N application [11]. Thus, the combination of N-efficient genotype development with the best management practices is therefore an important path for various stressed ecosystems around the world. It has often been shown that rice NUE, which integrates physiological and soil N supply capacities, decreases with increasing N supply in the soil [24].

To identify the appropriate breeding strategies, the germplasm must be evaluated for physiological variability in NUE [25], genotype interaction with N inputs, and different levels of N based on precise selection. Therefore, in the present study, we assessed the response of rice genotypes with different levels of N for several rice genotypes, where rice was fertilized with neem-oil-coated urea according to the regulatory requirements of India. Our experimental trials were based on the new idea of screening rice genotypes for a higher NUE. The main objectives were to: (i) evaluate the growth and yield components of rice genotypes under control versus half and recommended N supplies; (ii) investigate the differences between rice cultivars in terms of economic yield and harvest index; (iii) screen the rice genotypes based on the grain yield efficiency index.

## 2. Materials and Methods

### 2.1. Study Site and Soil Characteristics

A two-year field experiment (2020 and 2021) was carried out at the research farm of the ICAR (Indian Agricultural Research Institute, New Delhi, 28°38' N 77°10' E) during the Kharif (rainy) season. During the rice growing season, the annual maximum and minimum temperatures ranged from 20 °C to 31.6 °C and 4 °C to 28 °C, respectively. Before transplanting rice genotypes, undisturbed soil samples were collected from the upper 0.02 m soil profile. The physico-chemical properties of the soil are available in Supplementary Materials Table S1.

### 2.2. Experimental Design and Treatments

The experiments were designed in a split plot design with three levels of N in the main plot, viz. $N_0$ (control), $N_{60}$ ($\frac{1}{2}$ of the recommended N), $N_{120}$ (recommended N), and ten rice genotypes in sub-plots, viz. $G_1$ ('Tella Hamsa'), $G_2$ ('Vasumati'), $G_3$ ('VL Dhan 209'), $G_4$ ('Daya'), $G_5$ ('PB 1728'), $G_6$ ('Anjali'), $G_7$ ('Heera'), $G_8$ ('Birupa'), $G_9$ ('Nagina 22') and $G_{10}$ ('Nidhi'), replicated thrice. The characteristics and features of the rice genotypes are available in Supplementary Materials Table S2. Irrigation was performed using channels and hand-hoeing was performed to keep the crop weed-free. Nitrogen was supplied through the neem-oil-coated urea containing 46.6% N, and one ton of neem-coated urea contained 0.5 kg neem oil [26]. The N was applied in three splits: 1/2 at ten days after transplanting, 1/4 at tillering, and 1/4 at the panicle initiation stage. In addition to N, for the application of the recommended phosphorus and potassium, a one-time basal dose of single superphosphate and a muriate of potash was applied (Table 1).

**Table 1.** The timing and amount of N fertilization levels in two years' experimental trial.

| Treatment | Crop Stages for N Fertilization | | | Amount of N Fertilizer Applied through Neem-Oil-Coated Urea (46.6% N) | | | | SSP * (16% $P_2O_5$) | MOP ** (60% $K_2O$) |
|---|---|---|---|---|---|---|---|---|---|
| | Basal | Tillering | Panicle Initiation | Basal (Kg ha$^{-1}$) | Tillering (Kg ha$^{-1}$) | Panicle Initiation (Kg ha$^{-1}$) | Total N (Kg ha$^{-1}$) | Basal (Kg $P_2O_5$ ha$^{-1}$) | Basal (Kg $K_2O$ ha$^{-1}$) |
| $N_0$ | - | - | - | - | - | - | - | 60 | 40 |
| $N_{60}$ | 10 days after transplanting (50% of $N_{60}$) | 25% of $N_{60}$ | 25% of $N_{60}$ | 30 | 15 | 15 | 60 | 60 | 40 |
| $N_{120}$ | 10 days after transplanting (half of $N_{120}$) | 25% of $N_{120}$ | 25% of $N_{120}$ | 60 | 30 | 30 | 120 | 60 | 40 |

* SSP: single superphosphate; ** MOP: muriate of potash.

### 2.3. Sampling and Measurements

Four plants (hills) from each plot were randomly selected and cut to ground level at 30 and 60 DAT (days after transplanting), and physiological maturity for dry matter accumulation (DMA). The collected samples were sun-dried for 5–7 days and then oven-dried for 24 h at a temperature of 70 °C. Five rice plants were randomly selected from each plot for tiller counting at 30 DAT, 60 DAT and the harvesting stage. To determine the leaf area of the rice genotypes, the LICOR-3100 leaf area meter was used at 30 and 60 DAT. The chlorophyll content of leaves was measured indirectly using the SPAD 502 (Konica Minolta, Inc., Tokyo, Japan) chlorophyll meter. To ensure accuracy, fully expanded and youngest leaves were chosen, and 10 readings were taken from 10 different hills plant$^{-1}$ plot$^{-1}$ to represent the SPAD value. The readings were collected from the midpoint of the leaf blade, specifically between the leaf base and the tip. To obtain yield data, rice plants

were harvested from the central 4 m$^2$ region of each plot at maturity. The harvest index, 1000-grain weight, filled grains panicle$^{-1}$ (nos.) and panicles m$^{-2}$ were measured.

### 2.4. Calculation of Related Indicators

The grain yield efficiency index (GYEI) was used for classifying the genotypes as efficient or inefficient nitrogen users. Grain yield is the best measure of genotype evaluation in screening experiments. The GYEI helps to separate genotypes into high-yielding, stable, nutrient-efficient genotypes and low-yielding, unstable, nutrient-inefficient genotypes [27].

GYEI = (grain yield of rice genotype for a low-level N input/average grain yield of 10 rice genotypes for a low-level N input) × (grain yield of rice genotype for a high-level N input/average grain yield of rice genotypes for a high-level N input) [27].

### 2.5. Data Analysis

The data obtained from two consecutive years of the study were analyzed using the available, open-access software R studio (agricolae package of R Version), and a probability level of <0.05 was used to determine statistical significance. Two years of mean data of the crop traits recorded were used to make Pearson's correlation coefficient matrix or a diagram using the package 'Metan' (multi-environment trial analysis) in R studio [28]. The ANOVA details are provided in Supplementary Tables S3–S17.

## 3. Results

### 3.1. Growth

#### 3.1.1. Number of Tillers

Among all the genotypes, 'Nidhi' had the highest number of tillers at 30 and 60 days after transplanting (DAT) and at the harvesting stage, and the next best genotype was 'Daya' (Tables 2 and 3). In the first and second years at 30 DAT, the number of tillers m$^{-2}$ decreased significantly in the order of 'Nidhi' = 'PB 1728' = 'Daya' > 'Nagina 22' = 'Birupa' > 'Vasumati' > 'VL Dhan 209' > 'Tella Hamsa' = 'Heera' > 'Anjali'. All the genotypes responded up to N$_{120}$ and the highest numbers of tillers were observed at 60 DAT. During the first and second years, the order of genotypes with respect to the significantly decreasing tiller numbers at 60 DAT was 'Nidhi' > 'Daya' > 'PB 1728' > 'Nagina 22' > 'Birupa = 'Vasumati' > 'VL Dhan 209' = 'Tella Hamsa' > 'Heera' > 'Anjali', and 'Nidhi' = 'Birupa' = 'Daya' > 'PB 1728' > 'Vasumati' > 'Nagina 22' = 'VL Dhan 209' > 'Heera' = 'Tella Hamsa' > 'Anjali', respectively. Only four genotypes ('Daya', 'Heera', 'Nidhi', and 'Tella Hamsa') responded up to N$_{120}$ at the harvesting stage. The order of genotypes with respect to the significant decrease in the number of tillers at the harvesting stage was 'Nidhi' = 'Daya' = 'PB 1728' > 'Nagina 22' > 'Birupa' > 'Vasumati' = 'Tella Hamsa' > 'VL Dhan 209' = 'Heera' > 'Anjali' in the first season and 'Nidhi' > 'Birupa' > 'Daya' > 'PB 1728' > 'Vasumati' > 'Nagina 22' = 'Heera' = 'VL Dhan 209' $\geq$ 'Tella Hamsa' > 'Anjali' in the second season.

#### 3.1.2. Chlorophyll

SPAD (soil plant analysis development) values were measured in order to determine the status of chlorophyll in leaves at different growth stages. The genotype 'Daya' had a higher chlorophyll content at 30 and 60 DAT, which was on par with the genotype 'PB 1728' (Tables 2 and 3). With the N level N$_{120}$, the genotypes 'Birupa' and 'VL Dhan 209' reported the highest chlorophyll content. In both years at 30 DAT, the significantly decreasing chlorophyll content in different genotypes followed the order 'Daya' = 'PB 1728' = 'Vasumati' > 'Heera' > 'Tella Hamsa' = 'Nagina 22' > 'Anjali' > 'VL Dhan 209' > 'Nidhi' = 'Birupa'. Similarly the order of the genotypes with respect to the significantly decreasing chlorophyll content at 60 DAT was 'Daya' = 'PB 1728' = 'Vasumati' = 'Heera' > 'Tella Hamsa' > 'Nagina 22' > 'Anjali' > 'VL Dhan 209' > 'Nidhi' > 'Birupa' in the first year, and 'Daya' > 'PB 1728' > 'Vasumati' = 'Heera' > 'Tella Hamsa' > 'Nagina 22' > 'Anjali' > 'VL Dhan 209' > 'Nidhi' > 'Birupa' in the second year.

**Table 2.** Effect of nitrogen fertilization on tillers and chlorophyll index (30 DAT) of rice.

| Nitrogen/Genotype | SPAD Values at 30 DAT | | Tillers m$^{-2}$ at 30 DAT | |
|---|---|---|---|---|
| | 2020 | 2021 | 2020 | 2021 |
| $N_0$ | 25.3 [b] | 24.4 [b] | 89.0 [c] | 88.0 [c] |
| $N_{60}$ | 31.3 [a] | 30.4 [a] | 112 [b] | 111 [b] |
| $N_{120}$ | 33.5 [a] | 32.5 [a] | 129 [a] | 128 [a] |
| 'Tella Hamsa' | 31.1 [bc] | 30.2 [bc] | 103 [ef] | 102 [ef] |
| 'Vasumati' | 33.2 [ab] | 32.2 [ab] | 109 [cde] | 111 [bcd] |
| 'VL Dhan 209' | 27.6 [de] | 26.7 [de] | 107 [de] | 107 [de] |
| 'Daya' | 34.9 [a] | 34.0 [a] | 118 [ab] | 116 [abc] |
| 'PB 1728' | 33.3 [ab] | 32.5 [ab] | 116 [abc] | 113 [bcd] |
| 'Anjali' | 29.2 [cd] | 28.1 [cd] | 99 [f] | 99 [f] |
| 'Heera' | 31.1 [bc] | 30.3 [bc] | 102 [ef] | 103 [e] |
| 'Birupa' | 23.8 [f] | 22.6 [f] | 111 [bcd] | 118 [ab] |
| Nagina22 | 30.9 [bc] | 30.1 [bc] | 113 [bcd] | 109 [cde] |
| 'Nidhi' | 25.4 [ef] | 24.1 [ef] | 122 [a] | 122 [a] |
| N | 11.8 | 11.79 | 2.25 | 2.38 |
| V | 7.0 | 6.97 | 3.05 | 2.87 |
| N × V | ns | ns | ns | ns |

ns = non-significant; DAT = days after transplanting. Values in a column followed by different letters are significantly different at $p < 0.05$ as determined by LSD among the genotypes; Letters indicate the comparison among genotypes at different N levels; N = nitrogen, V = genotype.

**Table 3.** Effect of nitrogen × genotype interaction on chlorophyll index (SPAD) at 60 days after transplanting (DAT) and number of tillers at 60 DAT and harvesting stage of rice.

| Nitrogen × Genotype | | 'Tella Hamsa' | 'Vasumati' | 'VL Dhan 209' | 'Daya' | 'PB 1728' | 'Anjali' | 'Heera' | 'Birupa' | 'Nagina 22' | 'Nidhi' | Mean |
|---|---|---|---|---|---|---|---|---|---|---|---|---|
| | | | | | SPAD value (60 DAT) | | | | | | | |
| 2020 | $N_0$ | 38.7 [mn] | 42.3 [ij] | 33.6 [pq] | 40.9 [kl] | 43.9 [efgh] | 38.0 [n] | 42.6 [hij] | 24.6 [s] | 37.5 [n] | 34.8 [op] | 37.7 [c] |
| | $N_{60}$ | 44.9 [de] | 42.9 [ghi] | 38.8 [mn] | 46.5 [ab] | 45.1 [cde] | 41.4 [jk] | 44.0 [efg] | 30.3 [r] | 39.9 [lm] | 36.0 [o] | 41.0 [b] |
| | $N_{120}$ | 46.3 [bc] | 47.1 [ab] | 46.1 [bcd] | 47.8 [a] | 44.9 [de] | 43.5 [fghi] | 44.7 [ef] | 32.7 [q] | 46.3 [bc] | 38.8 [mn] | 43.8 [a] |
| | Mean | 43.3 [d] | 44.1 [bc] | 39.5 [f] | 45.1 [a] | 44.6 [ab] | 40.9 [e] | 43.8 [cd] | 29.2 [h] | 41.2 [e] | 36.5 [g] | |
| | | | | | * N × G = 1.32/* G × N = 1.30 | | | | | | | |
| 2021 | $N_0$ | 38.3 [p] | 41.4 [l] | 32.6 [u] | 40.0 [n] | 43.0 [i] | 37.0 [q] | 41.6 [kl] | 23.7 [x] | 36.5 [r] | 33.6 [t] | 36.8 [c] |
| | $N_{60}$ | 43.7 [h] | 41.8 [jp] | 38.4 [p] | 45.6 [cd] | 44.1 [g] | 40.0 [m] | 43.2 [i] | 29.4 [w] | 39.0 [o] | 35.0 [s] | 40.1 [b] |
| | $N_{120}$ | 45.3 [d] | 46.1 [b] | 44.8 [e] | 46.7 [a] | 44.5 [ef] | 42.0 [j] | 44.1 [fg] | 31.1 [v] | 45.9 [bc] | 37.3 [q] | 42.8 [a] |
| | Mean | 42.4 [d] | 43.1 [c] | 38.6 [g] | 44.1 [a] | 43.9 [b] | 40.0 [f] | 43.0 [c] | 28.1 [i] | 40.4 [e] | 35.3 [h] | |
| | | | | | * N × V = 0.38/* V × N = 0.37 | | | | | | | |
| | | | | | Tillers (60 DAT) | | | | | | | |
| 2020 | $N_0$ | 292 [op] | 307 [no] | 280 [pq] | 395 [k] | 399 [jk] | 239 [r] | 268 [q] | 311 [n] | 368 [m] | 413 [j] | 327 [c] |
| | $N_{60}$ | 365 [m] | 385 [kl] | 378 [lm] | 538 [e] | 504 [f] | 297 [nop] | 361 [m] | 390 [kl] | 501 [f] | 564 [d] | 428 [b] |
| | $N_{120}$ | 454 [h] | 474 [g] | 451 [h] | 620 [b] | 590 [c] | 375 [lm] | 433 [i] | 464 [gh] | 583 [c] | 647 [a] | 509 [a] |
| | Mean | 370 [f] | 388 [e] | 369 [f] | 517 [b] | 498 [c] | 304 [h] | 354 [g] | 388 [e] | 484 [d] | 541 [a] | |
| | | | | | * N × G = 16.94/* G × N = 16.21 | | | | | | | |
| 2021 | $N_0$ | 268 [t] | 310 [q] | 280. [st] | 399 [jk] | 368 [nop] | 238 [u] | 292 [rs] | 393 [kl] | 305 [qr] | 411 [j] | 326 [c] |
| | $N_{60}$ | 361 [p] | 389 [klm] | 378 [lmno] | 505 [f] | 501 [f] | 295 [qrs] | 365 [op] | 536 [e] | 383 [klmn] | 562 [d] | 428 [b] |
| | $N_{120}$ | 432 [i] | 464 [gh] | 450 [h] | 589 [c] | 582 [c] | 375 [mnop] | 453 [h] | 620 [b] | 473 [g] | 647 [a] | 508 [a] |
| | Mean | 354 [g] | 387 [e] | 369 [f] | 4976 [c] | 484 [d] | 303 [h] | 370 [f] | 516 [b] | 387 [e] | 540 [a] | |
| | | | | | Tillers at harvest | | | | | | | |
| 2020 | $N_0$ | 269 [pqr] | 302 [nop] | 277 [opq] | 392 [fghi] | 375 [ghij] | 238 [r] | 251 [qr] | 320 [lmn] | 379 [ghij] | 409 [fg] | 321 [c] |
| | $N_{60}$ | 361 [hijk] | 375 [ghij] | 346 [jkl] | 509 [cd] | 489 [de] | 303 [nop] | 340 [klm] | 396 [fgh] | 461 [e] | 530 [c] | 411 [b] |
| | $N_{120}$ | 408 [fg] | 391 [fghi] | 359 [ijk] | 595 [ab] | 569 [b] | 309 [mno] | 381 [ghij] | 416 [f] | 492 [de] | 621 [a] | 454 [a] |
| | Mean | 346 [fg] | 356 [f] | 328 [gh] | 499 [b] | 478 [c] | 284 [i] | 324 [h] | 377 [e] | 444 [d] | 520 [a] | |
| | | | | | * N × G = 34.9/* G × N = 34.5 | | | | | | | |
| 2021 | $N_0$ | 250 [qr] | 319 [lmn] | 276 [opq] | 374 [ghij] | 378 [ghij] | 238 [r] | 268 [pqr] | 391 [fghi] | 301 [nop] | 408 [fg] | 320 [c] |
| | $N_{60}$ | 338 [klm] | 395 [fgh] | 346 [jkl] | 488 [de] | 460 [e] | 303 [nop] | 360 [hijk] | 508 [cd] | 374 [ghij] | 529 [c] | 410 [b] |
| | $N_{120}$ | 380 [ghij] | 415 [f] | 358 [ijk] | 568 [b] | 492 [de] | 308 [mno] | 407 [fg] | 593 [ab] | 391 [fghi] | 619 [a] | 453 [a] |
| | Mean | 323 [h] | 376 [e] | 327 [gh] | 477 [c] | 443 [d] | 283 [i] | 345 [fg] | 497 [b] | 355 [f] | 519 [a] | |
| | | | | | * N × G = 34.6/* G × N = 34.2 | | | | | | | |

* LSD ($p = 0.05$) for nitrogen means at same or different level of genotypes; * LSD ($p = 0.05$) for genotypes means at same or different level of nitrogen. DAT = days after transplanting; SPAD = soil plant analysis development. Values in a column followed by different letters are significantly different at $p < 0.05$ as determined by LSD. Letters indicate the comparison among genotypes at different N levels.

### 3.1.3. Dry Matter Accumulation

In the first and second years at 30 days after transplanting (DAT), the dry matter accumulation (DMA) at 30 DAT, 60 DAT and the harvesting stage was 31.1% and 42.4%, 41% and 46.4%, and 32.9% and 38% higher with $N_{60}$ and $N_{120}$, respectively, over $N_0$ (Table 4). The highest DMA was produced by the 'Nidhi' and 'Birupa' genotypes at all the stages of growth. The order of genotypes with respect to the significantly decreasing DMA at 30 DAT was 'Nidhi' > 'PB 1728' > 'Daya' > 'Nagina 22' > 'Birupa' > 'Vasumati' > 'VL Dhan 209' > 'Tella Hamsa' > 'Anjali' = 'Heera' in 2020, and 'Birupa' > 'Nidhi' = 'Daya' > 'Vasumati' > 'PB 1728'> 'Nagina 22' = 'VL Dhan 209' > 'Heera' > 'Tella Hamsa' = 'Anjali' in 2021. Regarding the DMA at 60 DAT, only two genotypes, 'VL Dhan 209' and 'Daya', responded up to $N_{120}$ in the first year, but in the second year all the genotypes responded up to $N_{120}$. The order of the genotypes with respect to the significantly decreasing DMA at 60 DAT, in the first and second years, was 'Nidhi' = 'PB 1728' = 'Daya' > 'Birupa' = 'Nagina 22' > 'Vasumati' > 'VL Dhan 209' = 'Tella Hamsa' > 'Anjali' = 'Heera', and 'Birupa' > 'Nidhi' > 'Daya' > 'Vasumati' > 'PB 1728' > 'Nagina 22' > 'VL Dhan 209' > 'Heera' > 'Tella Hamsa' = 'Anjali', respectively. Similarly, the order of the genotypes with respect to the significantly decreasing DMA at the harvesting stage was 'Nidhi' > 'PB 1728' = 'Daya' > 'Birupa' = 'Nagina 22' > 'Vasumati' > 'VL Dhan 209' > 'Tella Hamsa' > 'Anjali' = 'Heera' in the first year, and 'Birupa' > 'Nidhi' > 'Daya' > 'Vasumati' > 'PB 1728' > 'Nagina 22' > 'VL Dhan 209' > 'Heera' > 'Tella Hamsa' = 'Anjali' in the second year.

**Table 4.** Effect of nitrogen × genotype interaction on dry matter accumulation (g m$^{-2}$) at 30, 60 DAT (days after transplanting) and at harvest of rice.

| Nitrogen × Genotype | | 'Tella Hamsa' | 'Vasumati' | 'VL Dhan 209' | 'Daya' | 'PB 1728' | 'Anjali' | 'Heera' | 'Birupa' | 'Nagina 22' | 'Nidhi' | Mean |
|---|---|---|---|---|---|---|---|---|---|---|---|---|
| 30 DAT (2020) | $N_0$ | 88 [no] | 104 [m] | 103 [m] | 117 [ijk] | 115 [jkl] | 84 [o] | 92 [n] | 114 [kl] | 104 [m] | 119 [ijk] | 104 [c] |
| | $N_{60}$ | 123 [hi] | 131 [fg] | 121 [hij] | 152 [d] | 160 [c] | 118 [ijk] | 108 [lm] | 134 [ed] | 149 [d] | 166 [bc] | 136 [b] |
| | $N_{120}$ | 132 [fg] | 140 [e] | 136 [ef] | 164 [c] | 172 [ab] | 126 [gh] | 121 [hij] | 149 [d] | 160 [c] | 178 [a] | 148 [a] |
| | Mean | 114 [h] | 125 [f] | 120 [g] | 145 [c] | 149 [b] | 109 [i] | 107 [i] | 132 [e] | 137 [d] | 154 [a] | |
| | | | | | * N × G = 7.1/* G × N = 7.8 | | | | | | | |
| 30 DAT (2021) | $N_0$ | 84 [n] | 106 [lm] | 102 [m] | 116 [ij] | 113 [jkl] | 91 [n] | 87 [n] | 118 [ij] | 100 [m] | 113 [jk] | 103 [c] |
| | $N_{60}$ | 117 [ij] | 148 [d] | 121 [hi] | 151 [d] | 133 [ef] | 107 [klm] | 122 [hi] | 165 [bc] | 130 [fg] | 159 [c] | 135 [b] |
| | $N_{120}$ | 125 [gh] | 159 [c] | 135 [ef] | 163 [bc] | 149 [d] | 120 [hij] | 131 [fg] | 177 [a] | 140 [e] | 170 [ab] | 147 [a] |
| | Mean | 109 [g] | 138 [c] | 120 [e] | 144 [b] | 131 [d] | 106 [g] | 114 [f] | 153 [a] | 123 [e] | 147 [b] | |
| | | | | | * N × G = 7.3/* G × N = 7.7 | | | | | | | |
| 60 DAT (2020) | $N_0$ | 311 [l] | 456 [hij] | 331 [l] | 578 [e] | 579 [e] | 245 [m] | 239 [m] | 489 [ghi] | 447 [ijk] | 588 [e] | 426 [c] |
| | $N_{60}$ | 473 [ghi] | 561 [ef] | 498 [gh] | 763 [bc] | 773 [ab] | 399 [k] | 396 [k] | 673 [d] | 689 [d] | 783 [ab] | 601 [b] |
| | $N_{120}$ | 491 [ghi] | 582 [e] | 523 [fg] | 789 [ab] | 800 [ab] | 414 [jk] | 408 [jk] | 697 [d] | 716 [cd] | 820 [a] | 624 [a] |
| | Mean | 425 [d] | 533 [c] | 451 [d] | 710 [a] | 717 [a] | 352 [e] | 348 [e] | 620 [b] | 617 [b] | 731 [a] | |
| | | | | | * N × G = ns/* G × N = ns | | | | | | | |
| 60 DAT (2021) | $N_0$ | 245 [v] | 514 [m] | 330 [t] | 577 [k] | 488 [o] | 238 [v] | 310 [u] | 587 [j] | 387 [s] | 577 [k] | 425 [c] |
| | $N_{60}$ | 397 [r] | 688 [h] | 498 [n] | 763 [e] | 672 [i] | 395 [rs] | 472 [p] | 782 [c] | 560 [l] | 772 [d] | 600 [b] |
| | $N_{120}$ | 413 [q] | 715 [f] | 522 [m] | 788 [c] | 697 [g] | 407 [q] | 491 [no] | 819 [a] | 582 [jk] | 799 [b] | 623 [a] |
| | Mean | 352 [i] | 639 [d] | 450 [g] | 709 [c] | 619 [e] | 347 [i] | 424 [h] | 730 [a] | 510 [f] | 716 [b] | |
| | | | | | * N × G = 8.5/* G × N = 8.2 | | | | | | | |
| Harvest (2020) | $N_0$ | 523 [l] | 692 [i] | 552 [kl] | 862 [f] | 860 [f] | 444 [m] | 439 [m] | 788 [gh] | 699 [i] | 923 [e] | 678 [c] |
| | $N_{60}$ | 709 [i] | 841 [fg] | 748 [hi] | 1145 [b] | 1160 [b] | 597 [jk] | 594 [jk] | 1009 [d] | 1033 [cd] | 1175 [ab] | 901 [b] |
| | $N_{120}$ | 737 [hi] | 873 [ef] | 784 [gh] | 1183 [ab] | 1100 [ab] | 620 [j] | 611 [jk] | 1046 [cd] | 1073 [c] | 1230 [a] | 936 [a] |
| | Mean | 656 [f] | 802 [d] | 695 [e] | 1063 [b] | 1073 [b] | 554 [g] | 548 [g] | 948 [c] | 935 [c] | 1109 [a] | |
| | | | | | * N × G = 60.1/* G × N = 60.4 | | | | | | | |
| Harvest (2021) | $N_0$ | 444 [v] | 769 [o] | 551 [t] | 862 [kl] | 787 [n] | 438 [v] | 522 [u] | 922 [j] | 620 [r] | 858 [l] | 677 [c] |
| | $N_{60}$ | 596 [s] | 1032 [h] | 747 [p] | 1144 [e] | 1008 [i] | 593 [s] | 708 [q] | 1174 [c] | 840 [m] | 1159 [d] | 900 [b] |
| | $N_{120}$ | 619 [r] | 1072 [f] | 783 [n] | 1182 [c] | 1045 [g] | 610 [r] | 736 [p] | 1228 [a] | 873 [k] | 1198 [b] | 935 [a] |
| | Mean | 553 [i] | 958 [d] | 694 [g] | 1062 [c] | 947 [e] | 547 [i] | 655 [h] | 1108 [a] | 778 [f] | 1072 [b] | |
| | | | | | * N × G = 12.1/* G × N = 24.5 | | | | | | | |

* LSD ($p = 0.05$) for nitrogen means at same or different level of genotypes; * LSD ($p = 0.05$) for genotypes means at same or different level of nitrogen; ns = non-significant; DAT = days after transplanting. Values in a column followed by different letters are significantly different at $p < 0.05$ as determined by LSD. Letters indicate the comparison among genotypes at different N levels.

### 3.1.4. Leaf Area Index

During the two years of the study, the genotype 'Nidhi' recorded the highest leaf area index (LAI) at 30 days after transplanting (DAT) and 60 DAT (Table 5). With respect to LAI, all the genotypes responded up to $N_{120}$ at 30 DAT, but at 60 DAT, the genotypes

had responses on par with $N_{60}$ and $N_{120}$. The N response order of the genotypes to the significantly decreasing LAI at 30 DAT was 'Nidhi' = 'Daya' > 'PB 1728' = 'Nagina 22' > 'Birupa' > 'Vasumati' > 'Tella Hamsa' > 'VL Dhan 209' > 'Heera' > 'Anjali' in the first year, and 'Nidhi' = 'Birupa' > 'Daya' = 'PB 1728' > 'Vasumati' > 'Nagina 22' > 'Heera' > 'VL Dhan 209' > 'Anjali' > 'Tella Hamsa' in the second year. With the exception of 'Anjali', 'Nidhi' and 'Tella Hamsa', all the genotypes responded up to $N_{120}$ after 60 DAT. The response of the genotypes to N with respect to LAI at 60 DAT was 'Nidhi' > 'Daya' > 'PB 1728' > 'Nagina 22' > 'Birupa' > 'Vasumati' > 'Tella Hamsa' > 'Daya' > 'Heera' > 'Anjali' in the first year, and 'Nidhi' > 'Birupa' > 'PB 1728' > 'Daya' > 'Vasumati' > 'Nagina 22' > 'Heera' > 'VL Dhan 209' > 'Anjali' > 'Tella Hamsa' in the second year.

**Table 5.** Effect of nitrogen × genotype interaction on leaf area index and number of panicles of rice.

| Nitrogen × Genotype | | 'Tella Hamsa' | 'Vasumati' | 'VL Dhan 209' | 'Daya' | 'PB 1728' | 'Anjali' | 'Heera' | 'Birupa' | 'Nagina 22' | 'Nidhi' | Mean |
|---|---|---|---|---|---|---|---|---|---|---|---|---|
| | | | | | LAI (30) | | | | | | | |
| 2020 | $N_0$ | 0.44 [r] | 0.51 [q] | 0.32 [s] | 0.78 [lmn] | 0.72 [no] | 0.16 [t] | 0.21 [t] | 0.62 [P] | 0.67 [op] | 0.81 [lm] | 0.52 [c] |
| | $N_{60}$ | 1.12 [j] | 1.21 [i] | 1.00 [k] | 1.66 [de] | 1.50 [f] | 0.77 [mn] | 0.84 [l] | 1.36 [h] | 1.49 [fg] | 1.69 [d] | 1.26 [b] |
| | $N_{120}$ | 1.54 [f] | 1.62 [e] | 1.42 [g] | 2.06 [a] | 1.91 [b] | 1.18 [ij] | 1.25 [i] | 1.76 [c] | 1.89 [b] | 2.11 [a] | 1.67 [a] |
| | Mean | 1.03 [e] | 1.11 [d] | 0.91 [f] | 1.5 [a] | 1.37 [b] | 0.7 [h] | 0.77 [g] | 1.25 [c] | 1.35 [b] | 1.54 [a] | |
| | | | | | * N × G = 0.07/* V × N = 0.11 | | | | | | | |
| 2021 | $N_0$ | 0.15 [r] | 0.60 [o] | 0.31 [q] | 0.66 [no] | 0.70 [mn] | 0.19 [r] | 0.42 [P] | 0.77 [lm] | 0.49 [P] | 0.80 [l] | 0.51 [c] |
| | $N_{60}$ | 0.76 [lm] | 1.34 [h] | 0.99 [k] | 1.47 [fg] | 1.48 [f] | 0.83 [l] | 1.11 [j] | 1.64 [de] | 1.20 [i] | 1.68 [cd] | 1.25 [b] |
| | $N_{120}$ | 1.17 [ij] | 1.75 [c] | 1.40 [gh] | 1.87 [b] | 1.89 [b] | 1.24 [i] | 1.52 [f] | 2.05 [a] | 1.61 [e] | 2.09 [a] | 1.66 [a] |
| | Mean | 0.69 [h] | 1.23 [c] | 0.9 [f] | 1.33 [b] | 1.36 [b] | 0.75 [g] | 1.01 [e] | 1.49 [a] | 1.1 [d] | 1.52 [a] | |
| | | | | | * N × G = 0.07/* G × N = 0.12 | | | | | | | |
| | | | | | LAI (60) | | | | | | | |
| 2020 | $N_0$ | 3.07 [r] | 3.27 [q] | 2.71 [s] | 3.66 [n] | 3.57 [no] | 2.06 [u] | 2.42 [t] | 3.37 [Pq] | 3.46 [op] | 3.86 [m] | 3.15 [b] |
| | $N_{60}$ | 4.66 [j] | 4.96 [i] | 4.36 [k] | 5.97 [b] | 5.67 [e] | 4.07 [l] | 4.17 [l] | 5.17 [h] | 5.47f | 6.20 [a] | 5.08 [a] |
| | $N_{120}$ | 5.02 [i] | 5.32 [g] | 4.98 [i] | 6.11 [b] | 5.91 [cd] | 4.13 [l] | 4.58 [j] | 5.41 [fg] | 5.83 [d] | 6.36 [a] | 5.35 [a] |
| | Mean | 4.25 [g] | 4.52 [f] | 4.02 [h] | 5.25 [b] | 5.05 [c] | 3.42 [j] | 3.72 [i] | 4.65 [e] | 4.92 [d] | 5.47 [a] | |
| | | | | | * N × G = 0.13/* G × N = 0.34 | | | | | | | |
| 2021 | $N_0$ | 2.05 [u] | 3.35 [Pq] | 2.70 [s] | 3.45 [op] | 3.55 [no] | 2.40 [t] | 3.05 [r] | 3.65 [n] | 3.25 [q] | 3.85 [m] | 3.13 [b] |
| | $N_{60}$ | 4.05 [l] | 5.15 [h] | 4.35 [k] | 5.45 [f] | 5.65 [e] | 4.15 [l] | 4.65 [j] | 5.95 [c] | 4.95 [i] | 6.20 [a] | 5.07 [a] |
| | $N_{120}$ | 4.10 [l] | 5.40 [fg] | 4.95 [i] | 5.80 [d] | 5.90 [cd] | 4.55 [j] | 5.00 [i] | 6.10 [b] | 5.30 [g] | 6.35 [a] | 5.33 [a] |
| | Mean | 3.4 [j] | 4.63 [e] | 4 [h] | 4.9 [d] | 5.03 [c] | 3.7 [i] | 4.23 [g] | 5.23 [b] | 4.5 [f] | 5.47 [a] | |
| | | | | | * N × G = 0.14/* G × N = 0.35 | | | | | | | |
| | | | | | Panicles m$^{-2}$ | | | | | | | |
| 2020 | $N_0$ | 213 [P] | 293 [n] | 246 [o] | 363 [fghi] | 297 [n] | 228 [op] | 236 [op] | 366 [fghi] | 249 [o] | 385 [ef] | 288 [c] |
| | $N_{60}$ | 302 [mn] | 379 [efgh] | 334 [jkl] | 428 [d] | 384 [efg] | 315 [lmn] | 317 [lmn] | 436 [d] | 341 [ijkl] | 463.96 [c] | 370 [b] |
| | $N_{120}$ | 326 [klm] | 395 [e] | 352 [hijk] | 478 [bc] | 491 [ab] | 335 [jkl] | 331 [jkl] | 492 [ab] | 357 [ghij] | 511 [a] | 407 [a] |
| | Mean | 281 [f] | 356 [d] | 311 [e] | 423 [b] | 391 [c] | 293 [f] | 294 [f] | 431 [b] | 316 [e] | 453 [a] | |
| | | | | | * N × G = 26.9/* G × N = 30.8 | | | | | | | |
| 2021 | $N_0$ | 213 [o] | 294 [m] | 246 [n] | 363 [fgh] | 297 [n] | 228 [no] | 236 [no] | 366 [fgh] | 249 [n] | 385 [ef] | 287 [c] |
| | $N_{60}$ | 302 [lm] | 379 [efg] | 334 [ijk] | 428 [d] | 384 [efg] | 315 [klm] | 317 [klm] | 436 [d] | 341 [hijk] | 464 [c] | 370 [b] |
| | $N_{120}$ | 325 [jkl] | 394 [e] | 351 [hij] | 477 [bc] | 491 [ab] | 335 [ijk] | 330 [jk] | 491 [ab] | 357 [ghi] | 510 [a] | 406 [a] |
| | Mean | 280 [f] | 356 [d] | 310 [e] | 423 [b] | 390 [c] | 292 [f] | 294 [f] | 431 [b] | 315 [e] | 453 [a] | |
| | | | | | * N × G = 26.9/* G × N = 30.8 | | | | | | | |

\* LSD ($p$ = 0.05) for nitrogen means at same or different level of genotypes; * LSD ($p$ = 0.05) for genotypes means at same or different level of nitrogen; DAT = days after transplanting. Values in a column followed by different letters are significantly different at $p < 0.05$ as determined by LSD. Letters indicate the comparison among genotypes at different N levels.

### 3.2. Yield-Attributing Characteristics

3.2.1. Number of Panicles

In both years of the study, the genotype 'Nidhi' had the highest number of panicles m$^{-2}$, and the next best genotype was 'Birupa' (Tables 5 and 6). Only four genotypes, viz. 'Daya', 'PB 1728', 'Birupa' and 'Nidhi', recorded a significant increase in panicles m$^{-2}$ up to $N_{120}$. The genotypes with significantly decreasing numbers of panicles m$^{-2}$ were ordered as follows: 'Nidhi' > 'Birupa' = 'Daya' > 'PB 1728' > 'Vasumati' > 'Nagina 22' = 'VL Dhan 209' > 'Tella Hamsa' = 'Heera' = 'Anjali'.

**Table 6.** Effect of nitrogen fertilization on yield attributes of rice crop.

| Treatment | Filled Grains Panicle$^{-1}$ | | 1000-Grain Weight (g) | |
|---|---|---|---|---|
| | **2020** | **2021** | **2020** | **2021** |
| $N_0$ | 92 [c] | 91 [c] | 20.5 [b] | 19.5 [b] |
| $N_{60}$ | 119 [b] | 118 [b] | 26.4 [ab] | 25.4 [ab] |
| $N_{120}$ | 128 [a] | 127 [a] | 29.6 [a] | 28.6 [a] |
| 'Tella Hamsa' | 96 [d] | 95 [d] | 24.0 [c] | 23.1 [cd] |
| 'Vasumati' | 115 [bc] | 114 [bc] | 25.5 [abc] | 24.5 [abcd] |
| 'VL Dhan209' | 102 [cd] | 102 [cd] | 24.9 [bc] | 24.1 [abcd] |
| 'Daya' | 129 [a] | 128 [a] | 27.3 [ab] | 26.3 [ab] |
| 'PB 1728' | 121 [ab] | 120 [ab] | 24.6 [bc] | 23.9 [abcd] |
| 'Anjali' | 98 [d] | 97 [d] | 23.6 [c] | 22.5 [d] |
| 'Heera' | 101 [d] | 100 [d] | 25.5 [abc] | 24.7 [abcd] |
| 'Birupa' | 131 [a] | 130 [a] | 27.9 [a] | 26.8 [a] |
| 'Nagina 22' | 105 [cd] | 104 [cd] | 24.4 [bc] | 23.6 [bcd] |
| 'Nidhi' | 133 [a] | 132 [a] | 27.0 [ab] | 25.8 [abc] |
| Interaction | ns | ns | ns | ns |

ns = non-significant; Values in a column followed by different letters are significantly different at $p < 0.05$ as determined by LSD. Letters indicate the comparison among genotypes at different N levels.

### 3.2.2. Filled Grains

The highest number of filled grains panicle$^{-1}$ was obtained by the genotype 'Nidhi', and it was on par with 'Birupa' and Daya' (Table 6). Genotypes with the significant decrease in the filled grains panicle$^{-1}$ followed the order: 'Nidhi' = 'Birupa' = 'Daya' = 'PB 1728' = 'Vasumati' > 'Nagina 22' = 'VL Dhan 209' > 'Tella Hamsa' = 'Anjali' = 'Heera'.

### 3.2.3. 1000-Grain Weight

Similar values of the 1000-grain weight were recorded with $N_{60}$ and $N_{120}$, but both of these N levels recorded significantly higher 1000-grain weights over the control, i.e., $N_0$ (Table 6). The genotypic difference in significantly decreasing 1000-grain weight followed the order 'Birupa' $\geq$ 'Daya' = 'Nidhi' = 'Heera' = 'Vasumati' $\geq$ 'VL Dhan 209' = 'PB 1728' = 'Nagina 22' $\geq$ 'Tella Hamsa' = 'Anjali'.

### 3.3. Grain Yield and Harvest Index

In both years, all the genotypes with $N_{120}$ application produced 8% and 2.7% higher grain yields and harvest indexes over $N_{60}$, and the genotypes 'Nidhi' and 'Daya' produced the highest grain yield and harvest index (Tables 7–9). The pooled analysis revealed a significant interaction effect between years, N levels, and genotypes on grain yield (GY). No significant interaction effect of years and N levels on GY was observed. The grain yields were statistically similar during the two years of the study. GY increased successively with increasing N levels up to $N_{120}$. The interaction effect of years and varieties on GY was significant, with only five genotypes ('Tela Hamsa', 'Daya', 'PB 1728', 'Nagina 22', and 'Nidhi') showing no significant differences between years. The genotypes 'Vasumati' and 'Anjali' produced significantly higher grain yields in the year 2020 as compared to 2021. Contrary to the above, the genotypes 'VL Dhan 209', 'Heera' and 'Birupa' recorded significantly higher grain yields during the year 2021 over the year 2020. 'Nidhi' had the highest GY at $N_{120}$, but in the second year, it was on par with 'Birupa'. The pooled results over two years showed that the 'Nidhi' genotypes had a significantly higher GY than all other genotypes. 'Daya' had the second-highest GY among all the genotypes and was significantly different from the others. 'Anjali' had the lowest GY and was significantly lower than all the other genotypes. 'Nidhi' recorded the highest yield at all N levels, indicating that it was the most efficient genotype for grain production across low, medium, and high N levels. The order of the genotypes with respect to the significantly decreasing grain yield was 'Nidhi' > 'Daya' > 'Birupa' $\geq$ 'PB 1728' > 'Vasumati' $\geq$ 'Nagina 22' >

'Tella Hamsa' ≥ 'VL Dhan 209' ≥ 'Heera' > 'Anjali'. The grain yield response (pooled) with $N_{120}$ application was only achieved in six genotypes: 'Vasumati' 'Daya', 'PB 1728', 'Birupa', 'Nagina 22', and 'Nidhi'. Regarding the significantly decreasing HI, the sequence of genotypes was 'Nidhi' > 'Daya' > 'Nagina22' > 'PB1728' = 'Birupa' = 'Vasumati' = 'Tella Hamsa' > 'Heera' ≥ 'Anjali' ≥ 'VL Dhan 209' in the first year, and 'Nidhi' ≥ 'Birupa' ≥ 'Daya' = 'VL Dhan 209' = 'Vasumati' = 'Nagina 22' ≥ 'Heera' = 'VL Dhan 209' = 'Anjali' = 'Tella Hamsa' in the second year.

**Table 7.** Grain yield of rice (t ha$^{-1}$) as influenced by years, nitrogen and genotypes (main effect table).

| Treatment | Grain Yield of Rice (t ha$^{-1}$) |
|---|---|
| Year | |
| 2020 | 3.86 [a] |
| 2021 | 3.77 [a] |
| Nitrogen | |
| $N_0$ | 2.82 [c] |
| $N_{60}$ | 4.15 [b] |
| $N_{120}$ | 4.48 [a] |
| Genotype | |
| 'Tella Hamsa' | 2.85 [e] |
| 'Vasumati' | 3.86 [d] |
| 'VL Dhan209' | 2.94 [e] |
| 'Daya' | 5.06 [b] |
| 'PB 1728' | 4.40 [c] |
| 'Anjali' | 2.47 [f] |
| 'Heera' | 2.84 [e] |
| 'Birupa' | 4.40 [c] |
| 'Nagina 22' | 3.81 [d] |
| 'Nidhi' | 5.54 [a] |

Values in a column followed by different letters are significantly different at *p* < 0.05 as determined by LSD. Letters indicate the comparison among years, nitrogen and genotypes.

**Table 8.** Effect of years × nitrogen × genotype interaction on grain yield (t ha$^{-1}$) of rice.

| Year/Genotype | 2020 | | | | 2021 | | | |
|---|---|---|---|---|---|---|---|---|
| | $N_0$ | $N_{60}$ | $N_{120}$ | Mean | $N_0$ | $N_{60}$ | $N_{120}$ | Mean |
| 'Tella Hamsa' | 2.80 [kl] | 3.60 [gh] | 3.80 [fg] | 3.40 [f] | 1.93 [o] | 2.81 [kl] | 2.19 [mno] | 2.31 [f] |
| 'Vasumati' | 2.60 [lm] | 3.90 [fg] | 4.05 [f] | 3.52 [ef] | 3.10 [jk] | 4.35 [f] | 5.15 [e] | 4.20 [c] |
| 'VL Dhan209' | 2.40 [mn] | 3.20 [ij] | 3.15 [ijk] | 2.92 [g] | 2.15 [no] | 3.45 [ghij] | 3.28 [hij] | 2.96 [e] |
| 'Daya' | 3.60 [gh] | 5.30 [c] | 6.60 [b] | 5.17 [b] | 3.45 [ghij] | 5.33 [de] | 6.06 [abc] | 4.95 [b] |
| 'PB 1728' | 3.40 [hi] | 4.90 [d] | 5.30 [c] | 4.53 [c] | 3.14i [jk] | 4.53 [f] | 5.13 [e] | 4.27 [c] |
| 'Anjali' | 1.95 [o] | 2.90 [jkl] | 3.05 [ijk] | 2.63 [f] | 2.01 [no] | 2.62 [lm] | 2.28 [mno] | 2.30 [h] |
| 'Heera' | 2.20 [no] | 2.90 [jkl] | 3.00 [jk] | 2.70 [gh] | 2.05 [no] | 3.32 [hij] | 3.56 [ghi] | 2.98 [e] |
| 'Birupa' | 3.10 [ijk] | 3.75 [fgh] | 4.05 [f] | 3.63 [e] | 3.38 [ghij] | 5.62 [cd] | 6.48 [a] | 5.16 [ab] |
| 'Nagina 22' | 3.15 [ijk] | 4.45 [e] | 5.20 [cd] | 4.27 [d] | 2.42 [lmn] | 3.82 [g] | 3.79 [g] | 3.34 [d] |
| 'Nidhi' | 3.80 [fg] | 6.25 [b] | 7.35 [a] | 5.80 [a] | 3.68 [gh] | 5.91 [bc] | 6.24 [ab] | 5.28 [a] |
| Mean | 2.90 [c] | 4.12 [b] | 4.56 [a] | | 2.73 [c] | 4.18 [b] | 4.41 [a] | |
| | * N:G:Y = 0.41/** N:G = 0.29/*** Y:G = 0.24/**** Y:N = 0.20 | | | | | | | |

N = nitrogen; G = genotype; Y = year; *, **, ***, **** = LSD (*p* = 0.05). Values in a column followed by different letters are significantly different at *p* < 0.05 as determined by LSD. Letters indicate the comparison among genotypes at different N levels.

**Table 9.** Effect of nitrogen × genotype interaction on harvest index (%) of rice.

| Nitrogen × Genotype | | 'Tella Hamsa' | 'Vasumati' | 'VL Dhan 209' | 'Daya' | 'PB 1728' | 'Anjali' | 'Heera' | 'Birupa' | 'Nagina 22' | 'Nidhi' | Mean |
|---|---|---|---|---|---|---|---|---|---|---|---|---|
| | | Harvest index (%) | | | | | | | | | | |
| 2020 | $N_0$ | 35.5 jkl | 36.5 ijk | 28.5 p | 40.5 ef | 36.0 ijk | 30.5 op | 32.5 mno | 35.5 jkl | 39.0 fgh | 44.0 bcd | 35.8 c |
| | $N_{60}$ | 36.5 ijk | 37.5 hij | 30.5 op | 43.0 cd | 37.5 hij | 31.5 no | 32.5 mno | 36.5 ijk | 40.0 efg | 45.5 ab | 37.1 b |
| | $N_{120}$ | 36.5i jk | 34.5 klm | 32.5 mno | 45.0 v | 38.0 ghi | 33.0 mn | 33.5 lmn | 37.5 hij | 42.0 de | 47.5 a | 38.0 a |
| Mean | Mean | 36.2 d | 36.2 d | 30.5 f | 42.8 b | 37.2 d | 31.7 ef | 32.8 e | 36.5 d | 40.3 c | 45.7 a | |
| | | * N × G = 2.08/* G × N = 2.08 | | | | | | | | | | |
| 2021 | $N_0$ | 37.3 ij | 39.7 efghij | 37.0 j | 41.2 cdefghi | 40.1 efghij | 37.6 hij | 37.9 hij | 42.0 bcdefg | 37.9 hij | 43.2 abcde | 39.4 c |
| | $N_{60}$ | 38.1 ghij | 40.3 defghij | 38.9 fghij | 42.6 bcdef | 41.4 bcdefgh | 38.4 ghij | 39.1 fghij | 43.2 abcde | 39.2 fghij | 45.3 ab | 40.7 b |
| | $N_{120}$ | 39.1 fghij | 40.9 cdefghij | 40.8 cdefghij | 44.2 abcd | 42.7 bcdef | 39.2 fghij | 40.3 defghij | 44.4 abc | 40.5 cdefghij | 46.8 a | 41.9 a |
| | Mean | 38.2 d | 40.0 cd | 38.9 d | 42.7 b | 41.4 bc | 38.4 d | 39.1 d | 43.2 ab | 39.2 cd | 45.1 a | |
| | | * N × G = ns/* G × N = ns | | | | | | | | | | |

* LSD ($p$ = 0.05) for nitrogen means at same or different level of genotypes; * LSD ($p$ = 0.05) for genotypes means at same or different level of nitrogen; ns = non-significant. Values in a column followed by different letters are significantly different at $p < 0.05$ as determined by LSD. Letters indicate the comparison among genotypes at different N levels.

### 3.4. Grain Yield Efficiency Index

Based on the grain yield efficiency index (GYEI), the lowland rice genotypes were classified as efficient (GYEI ≥ 1), moderately efficient (GYEI < 1–0.5), and inefficient (GYEI ≤ 0.5) with respect to the responses to $N_{60}$ and $N_{120}$ (Table 10, Figure 1). In this study, four rice genotypes ('Daya', 'PB 1728', 'Nagina 22', 'Nidhi') were classified as efficient N users, but in the second year, 'Vasumati' and 'Birupa' also became efficient N users and 'Nagina 22' became a moderately efficient N utilizer. Four genotypes ('Tella Hamsa', 'Vasumati', 'VL Dhan 209', 'Birupa') were grouped as moderately efficient N utilizers, and two genotypes ('Anjali', 'Heera') were inefficient N utilizers in 2020. In 2021, the genotype 'Tella Hamsa' was an inefficient N user.

**Table 10.** Grain yield efficiency index (GYEI) of 10 rice genotypes.

| Genotype | 2020 | 2021 |
|---|---|---|
| 'Tella Hamsa' | 0.73 | 0.45 |
| 'Vasumati' | 0.84 | 1.09 |
| 'VL Dhan 209' | 0.54 | 0.68 |
| 'Daya' | 1.87 | 1.63 |
| 'PB 1728' | 1.38 | 1.17 |
| 'Anjali' | 0.47 | 0.39 |
| 'Heera' | 0.46 | 0.63 |
| 'Birupa' | 0.81 | 1.81 |
| 'Nagina 22' | 1.24 | 0.84 |
| 'Nidhi' | 2.46 | 2.02 |

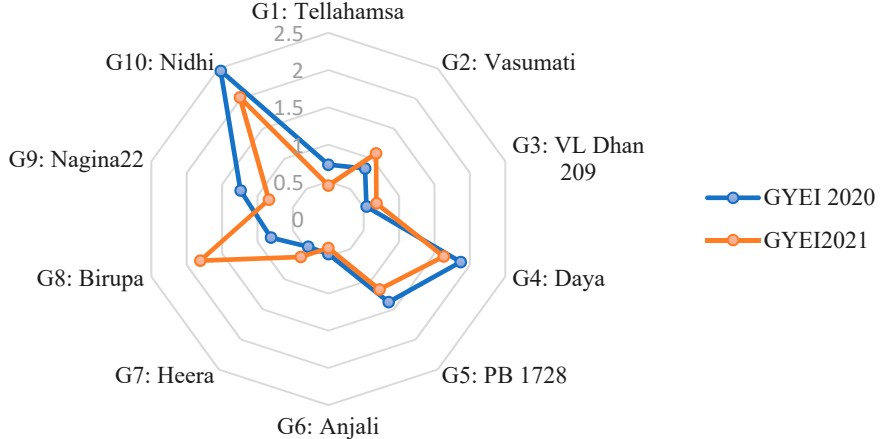

**Figure 1.** Pictorial representation using radar charts of mean grain yield efficiency index (GYEI) of ten rice genotypes grown over two years at different N levels during 2020–21.

### 3.5. Correlation Analysis among Agro-Morphological Traits or Parameters of Rice

The relationship among all plant traits such as leaf area index at 30 DAT (days after transplanting), 60 DAT (LAI-30 and LAI-60), chlorophyll index at 30 and 60 DAT (SPAD-30 and 60), tiller numbers at 30 and 60 DAT (TILL-30 and 60) and the harvesting stage (TILL-H), dry matter accumulation at 30 and 60 DAT (DMA-30 and 60) and harvest (DMA-H), panicles $m^{-2}$ (PAN), filled grains panicle$^{-1}$ (FGP), test weight (TW), grain yield (GY), straw yield (SY), biological yield (BY), and harvest index (HI) were analyzed using Pearson's correlation coefficient (Figures 2–4). With respect to different levels of $N_0$, $N_{60}$ and $N_{120}$, SPAD-30 was positively correlated with SPAD-60, but all remaining traits had a negative and non-significant correlation. Similarly, SPAD-60 showed a non-significant correlation with all traits. With the increase in N levels to $N_{60}$, the test weight showed a positive correlation with TILL-30 ($p < 0.01$) and PAN ($p < 0.05$), but with all remaining traits there was a negative, non-significant correlation. At the full dose of N application, the test weight showed a positive correlation with FGP, HI ($p < 0.05$), and TILL-30 ($p < 0.01$). The remaining agro-morphological traits or parameters of rice showed a significant positive correlation with each other at the three levels of N fertilization.

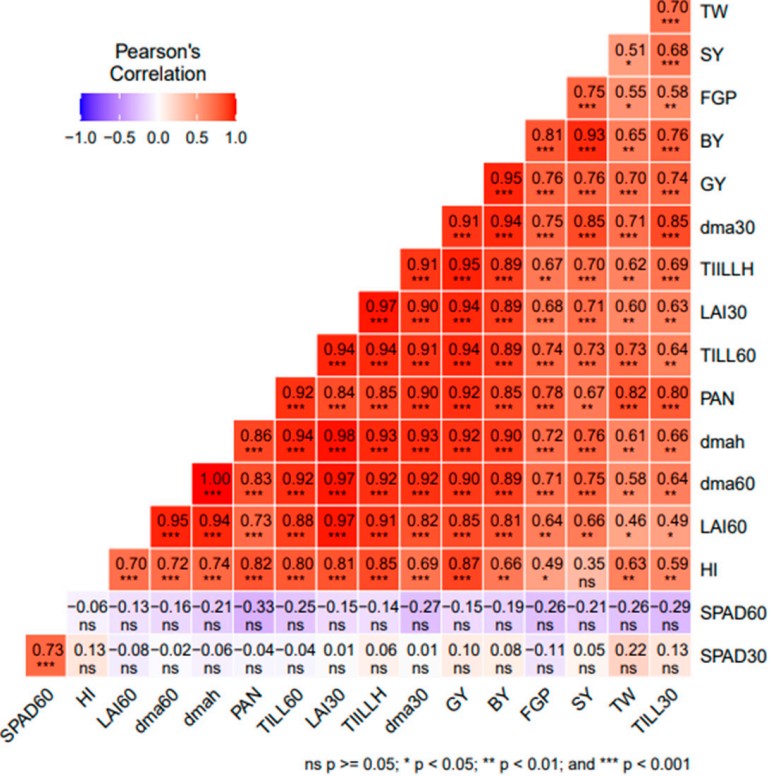

**Figure 2.** Pearson's correlation among agro-morphological traits or parameters of ten rice genotypes at the zero fertilization level ($N_0$). The correlation coefficient (R-value) was calculated from the mean of two-year data from 2020 to 2021. *** = significance at $p < 0.001$, ** = significance at $p < 0.01$ and * = $p < 0.05$, respectively. DAT = days after transplanting; leaf area index at 30 DAT and 60 DAT (LAI-30 and LAI-60); chlorophyll index at 30 and 60 DAT (SPAD-30 and 60); tiller numbers at 30 and 60 DAT (TILL-30 and 60) and harvesting stage (TILL-H); dry matter accumulation at 30 and 60 DAT (DMA-30 and 60) and harvest (DMA-H); panicles $m^{-2}$ (PAN); filled grains panicle$^{-1}$ (FGP); test weight (TW); grain yield (GY); straw yield (SY); biological yield (BY); harvest index (HI).

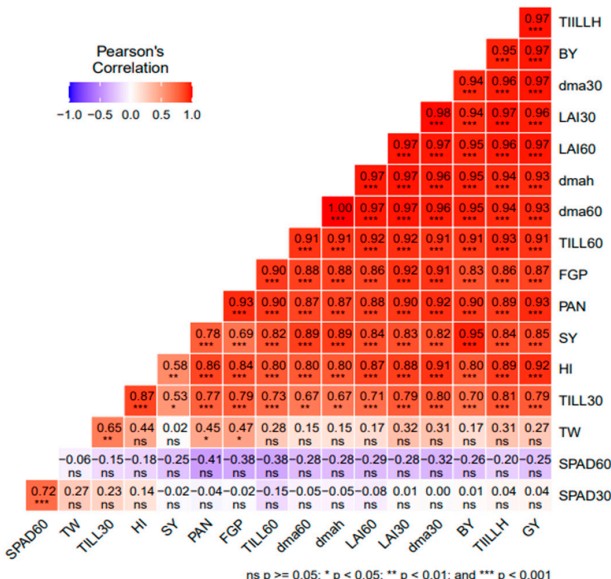

**Figure 3.** Pearson's correlation among agro-morphological traits or parameters of ten rice genotypes at half the recommended dose of nitrogen ($N_{60}$). The correlation coefficient (R-value) was calculated from the mean of two-year data from 2020 to 2021. *** = significance at $p < 0.001$, ** = significance at $p < 0.01$ and * = $p < 0.05$, respectively. DAT = days after transplanting; leaf area index at 30 DAT and 60 DAT (LAI-30 and LAI-60); chlorophyll index at 30 and 60 DAT (SPAD-30 and 60); tiller numbers at 30 and 60 DAT (TILL-30 and 60) and harvesting stage (TILL-H); dry matter accumulation at 30 and 60 DAT (DMA-30 and 60) and harvest (DMA-H); panicles $m^{-2}$ (PAN); filled grains panicle$^{-1}$ (FGP); test weight (TW); grain yield (GY); straw yield (SY); biological yield (BY); harvest index (HI).

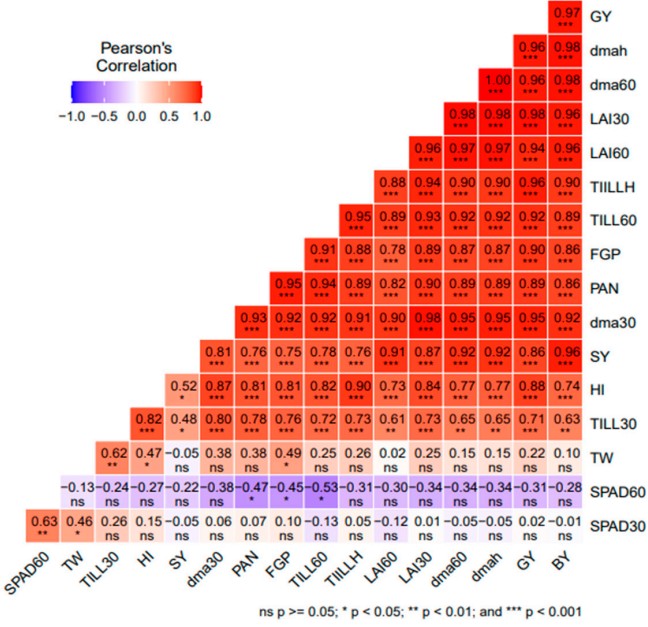

**Figure 4.** Pearson's correlation among agro-morphological traits or parameters of ten rice genotypes at the full dose of nitrogen ($N_{120}$). The correlation coefficient (R-value) was calculated from the mean of two-year data from 2020 to 2021. *** = significance at $p < 0.001$, ** = significance at $p < 0.01$ and * = $p < 0.05$, respectively. DAT = days after transplanting; leaf area index at 30 DAT and 60 DAT (LAI-30 and LAI-60); chlorophyll index at 30 and 60 DAT (SPAD-30 and 60); tiller numbers at 30 and 60 DAT (TILL-30 and 60) and harvesting stage (TILL-H); dry matter accumulation at 30 and 60 DAT (DMA-30 and 60) and harvest (DMA-H); panicles $m^{-2}$ (PAN); filled grains panicle$^{-1}$ (FGP); test weight (TW); grain yield (GY); straw yield (SY); biological yield (BY); harvest index (HI).

## 4. Discussion

Overall, the yearly variations in the significance of the data were quite similar, mainly due to the normal weather and identical field conditions across the two years. Though the grain yields were statistically similar during the two years of the study, the interaction of years, N levels and genotypes had a significant effect on the grain yield of rice.

### 4.1. Effect of Nitrogen

The application of N significantly increased all the agro-morphological performance measures including number of tillers, leaf area index (LAI), and chlorophyll content at all stages of growth. Conversely, each growth characteristic showed a different response to the application of N because each genotype had a unique genetic makeup. The better growth characteristics with increased N levels may be attributable to some genotypes being more N responsive and others being less N responsive, which may have reduced N loss and promoted better crop growth and development. The values recorded in the various performance indicators increased significantly from $N_0$ to $N_{60}$ at all stages of growth, but as the levels of N increased from $N_{60}$ to $N_{120}$, the data showed a significant difference between genotypes. The number of tillers $m^{-2}$, LAI and chlorophyll content showed a similar trend. As growth consistently increased with N input, the number of tillers steadily increased until reaching the peak at 60 DAT. With $N_{120}$ application, the genotypes showed a significant increase in tiller numbers compared to $N_{60}$ and $N_0$. This could have been the result of higher levels of LAI and chlorophyll in the leaf at all stages of growth, which improved the source-to-sink ratio and caused the plant to assimilate more photosynthates.

Nitrogen input regulates several plant hormones (e.g., auxin, cytokinin) and the expression of genes, all of which affect the emergence and growth of tillering [29–32]. Furthermore, Sui et al. [33] found that increasing the levels of N application at the recommended rate led to higher N absorption and utilization, which increased photosynthetic activity, so that N was quickly assimilated by rice plants to accelerate their growth. Generally, additional nitrogen considerably improved vegetative development, which ultimately led to better overall vegetative growth, including taller plants and more tillers $m^{-2}$. Different levels of N had a pronounced effect on the dry matter accumulation (DMA) of the rice genotype. Nitrogen fertilization in the first and second years with $N_{60}$ and $N_{120}$ increased the DMA at 30 days after transplanting (DAT) (30.8–31.4% and 42.2–42.5%), DMA-60 (40.9–4% and 46.3–46.4%) and DMA at harvest (32.8–32.9% and 37.9–38%) over $N_0$, respectively, which can be attributed to the split application of N in the genotypes and the maximum growth rate occurring at all these stages. A well-balanced application of N can increase the dry matter content by producing photo-assimilates in the leaves, which are the main center of plant growth during the vegetative stage, and later distributing the assimilates to the reproductive organs [34,35]. The yield attributes of different genotypes, such as the number of panicles $m^{-2}$, the number of filled grains panicle$^{-1}$ and the test weight, were significantly influenced by different levels of N.

The glutamate synthetase enzyme serves as a mobilizer of N during senescence and determines the number of grains, grain size, and grain filling in rice genotypes [36]. Nitrogen levels from $N_0$ to $N_{120}$ showed significant increases in panicles $m^{-2}$. This increase was possibly the reason behind the increase in tillers $m^{-2}$ with successive increases in N levels. In rice production, N is generally applied during the early vegetative stages to promote the number of panicles plant$^{-1}$. The application of N to rice plants in the early vegetative stages promotes panicles $m^{-2}$, while topdressing with N during the initiation stage of panicles increases the filled grains panicle$^{-1}$ [37]. Generally, greater tiller numbers $m^{-2}$ and panicle numbers are produced in genotypes that have higher N uptake [38]. The number of panicle-filled grains increased significantly at the N levels $N_{60}$ and $N_{120}$. The genotypes remained green for a longer time with increased N levels to some extent. This is applicable to genotypes based on genetic characteristics and their physiology. In both years, the application of $N_{60}$ and $N_{120}$ recorded the highest LAI, so this could have been the reason

behind the higher number of filled grains panicle$^{-1}$ and the higher levels of source–sink ratio, which resulted in high carbohydrate formation from high photosynthesis.

The grain yield and genotype harvest index are, in general, significantly and positively correlated with N levels, soil fertility, weather, and environmental condition of a given area. The average grain yield increased for N$_{60}$ and N$_{120}$ over the control by 42.1% and 57.2% in the first season and 53.1% and 61.5% in the second season. N$_{120}$ produced 10.7% and 5.5% higher economic yields of the rice genotypes over N$_{60}$ in 2020 and 2021, respectively. Higher levels of N led to greater uptake, which was correlated with a greater sink in the genotype. However, it differed from genotype to genotype because of the genetic potential. A possible strategy to use energy resources efficiently and improve yield performance is to use suitable N-efficient genotypes to increase the rice harvest index. Several studies reported that leaf photosynthetic capacity and grain yield were closely associated with N fertilization, and a deficiency of N decreased both the photosynthetic capacity and grain yield [39,40]. Therefore, the rice genotype efficiency of using N is directly and positively governed by the genetic makeup of the cultivar.

*4.2. Rice Genotype*

Rice genotypes with different growth parameters, such as tiller numbers m$^{-2}$, leaf area, and chlorophyll content, were significantly influenced by the genetic characteristics of the cultivar. The genotypes 'Nidhi' and 'Daya' were significantly superior in tiller numbers m$^{-2}$. In particular, the genotypes 'Nidhi' and 'PB 1728' showed the highest LAI, while genotypes 'Vasumati' and 'Daya' showed the highest chlorophyll content compared to the other rice genotypes. The genetic characteristics of the genotype may be the cause of the different growth behavior of rice cultivars [41,42]. The 'Nidhi' genotype showed a higher leaf area index due to a higher tiller number m$^{-2}$ and a longer leaf width of the genotype. Rice plant tiller numbers are influenced by LAI [43]. In rice genotypes, other factors such as plant height [44,45], panicle size [46], and hormones [47] may be responsible for differences in tillering capacity between rice genotypes.

Genotypes such as 'Nidhi' and 'Birupa' had higher yield attributes such as panicle number m$^{-2}$, filled grains panicle$^{-1}$, and test weight. The higher yield attributes of this genotype were due to the greater surface area of the rice roots, the better overall growth and development of the plant, and the greater photosynthetic efficiency of the leaf area index at flowering and physiological maturity. The genotypes 'Nidhi' and 'Daya' produced a significantly greater grain yield and harvest index than the other genotypes. The highest grain yield and harvest index in this genotype was due to the better formation of yield-attributing characteristics. The number of panicles m$^{-2}$ was directly associated with grain yield (Tables 6 and 7), also according to a previous study [48]. A higher yield is the result of an increased yield sink capacity, and the number of panicles per unit area is primarily responsible for the differential responses of grain yield to N rates [49–51]. Counce et al. [52] reported that rice grain yields increased mainly by increasing the tiller numbers. Similarly, in our study, the highest grain yield of 'Nidhi' and 'Daya' under the N$_{120}$ treatment was mainly attributed to the greater number of tillers, panicles m$^{-2}$, harvest index and increased biomass accumulation.

The grain yield efficiency index (GYEI) had a direct positive correlation with the grain yields of the genotypes. Grain yield is the best way to assess a genotype in screening experiments. Regarding GYEI, the four genotypes 'Nidhi', 'Daya', 'PB 1728', and 'Nagina 22' responded well to different levels of N, and these genotypes had a GYEI $\geq 1$ (Table 10). The genotypes 'Daya', 'PB 1728', 'Nagina 22' and 'Nidhi' were efficient N utilizers. Our results indicated that the efficiency of N use was different in all rice genotypes. There is a lot of information on how different cultivars and species of rice use nitrogen differently [53,54]. Plant characteristics such as the shape of the root system and the density of root hairs have been linked to differences in the plants' ability to thrive in low-nitrogen soils.

*4.3. Interaction of Years, Nitrogen and Genotypes*

The interaction of years, nitrogen levels and genotypes was found to have a significant effect on grain yield. The genotypes 'Tela Hamsa', 'Daya', 'PB 1728', 'Nagina 22', and 'Nidhi' showed no significant differences between years. These former genotypes produced similar yields during both years and can be considered stable genotypes. However, the grain yields of the genotypes 'Vasumati', 'Anjali', 'VL Dhan 209', 'Heera' and 'Birupa' differed significantly during the two years, and can thus be considered unstable genotypes. The pooled yield across the two years also showed that, in general, the most stable genotypes, viz. 'Nidhi', 'Daya' and 'PB 1728', produced significantly higher grain yields over the most unstable genotypes, viz., 'Anjali', 'VL Dhan 209' and 'Heera'. Grain yield is a quantitative parameter that is determined by the additive main effect of environment (E) and genotype (G) in addition to the non-additive effect of the G X E interaction (GEI) [55]. Breeders focus on the GEI effect to identify the yield stability of genotypes across different conditions and environments, which cannot be determined separately [56,57]. The GY heritability is exposed to variability across different environments [58], which hinders the accuracy of superior varietal selection processes [59]. Thus, widely adapted genotypes with the ability to produce stable, high yields across diversified environments constitute a major objective for rice breeders [60].

*4.4. Relationship between Agro-Morphological Traits or Parameters of Rice*

Our results showed that the chlorophyll index at 30 DAT (SPAD-30) and 60 DAT (SPAD-60), test weight (TW), and the response of other plant traits in various rice genotypes were dramatically different when the levels of N increased. With successive increases in N levels, all traits except SPAD-30, SPAD-60, and test weight (TW) were significantly and positively correlated with each other (see Figures 2–4) [61–63]. In the correlation analysis, panicles m$^{-2}$ (PAN) did not change with different levels of N. Harvest index had a positive correlation with GY [64]. Several studies reported that researchers could produce a high yield and nitrogen use efficiency (NUE) by increasing the distribution of biomass into the grain, rather than simply increasing its concentration [20]. On the basis of these results, it can be concluded that the application of low, medium, and high doses of N alone is not sufficient to improve the NUE, and that the selection of a genotype with the genetic potential for N accumulation is also an important factor for improving the NUE. Therefore, improving N metabolism and identifying genetic pathways involved in N metabolism may be the key to improving the NUE in rice genotypes.

**5. Conclusions**

The results obtained here are from field experiments conducted over two years that considered the overall performance of rice genotypes at different N levels and broadly based on multiple performance indicators. In particular, with respect to the grain yield, harvest index and grain yield efficiency index, the genotypes 'Nidhi' and 'Daya' can be considered N-efficient genotypes. Furthermore, with respect to the grain yield, the most stable genotypes were found to be 'Nidhi', 'Daya' and 'PB 1728', and the most unstable genotypes were 'Anjali', 'VL Dhan 209' and 'Heera'. Thus, the genotypes 'Nidhi', 'Daya' and 'PB 1728' could be used to enhance the nitrogen use efficiency and to breed nitrogen-efficient genotypes.

**Supplementary Materials:** The following supporting information can be downloaded at: https://www.mdpi.com/article/10.3390/su15118793/s1, Table S1: Initial Physico-chemical properties of soil in the experimental field; Table S2: Date of transplanting, date of maturity and harvesting, duration transplanting to maturity, duration nursery to maturity, seeding age, 50% flowering date and month; Table S3–S17: Analysis of variance for effect of nitrogen fertilization. References [65–68] are cited in the supplementary materials.

**Author Contributions:** Conceptualization, supervision, data curation, methodology, formal analysis, writing—original draft, S.G., D.K., Y.S.S., A.B., S.M., M.S.C., A.K. and R.K.; funding acquisition, writing—review and editing, A.H.P., N.R., H.P. and M.A.S. All authors have read and agreed to the published version of the manuscript.

**Funding:** This research was financially supported by the UKRI-GCRF South Asia Nitrogen Hub (SANH, NE/S009019/1).

**Institutional Review Board Statement:** Not applicable.

**Informed Consent Statement:** Not applicable.

**Data Availability Statement:** Not applicable.

**Acknowledgments:** This work was supported by the UKRI-GCRF South Asia Nitrogen Hub (SANH, NE/S009019/1). The authors thank the ICAR-Indian Agricultural Research Institute, New Delhi, for providing a senior research fellowship for the entire duration of Ph.D. work to S.G.

**Conflicts of Interest:** All authors of the manuscript reported no potential conflict of interest.

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
