# Peer review of "Field-Based Evaluation of Rice Genotypes for Enhanced Growth, Yield Attributes, Yield and Grain Yield Efficiency Index in Irrigated Lowlands of the Indo-Gangetic Plains"

_sustainability, doi:10.3390/su15118793_

Round 1
Reviewer 1 Report
The experimental trials are based on the novel concept of screening rice genotypes for higher NUE. The main objectives are, (a) to compare the growth and yield components of different rice genotypes when provided with low, half, and recommended levels of nitrogen (N) and (b) to examine the differences between different rice cultivars in terms of economic yield and harvest index. This study is relevant to the field of interest.
Methodology sounds good. Include a mention of chlorophyll estimation in the methodology.
The text should be written in the same style. Nitrogen x Variety or Nitrogen x Genotype in Table 5 is the example.
Overall, there should be some light editing and proofreading of the grammar.
Reviewer 2 Report
Dear authors, you evaluated the rice genotypes for their NUE and observed plant growth attributes. The authors have done exhaustive work, I have the following points to comment;
1. In my opinion, the title is long and can be concise as - Evaluation of Rice Genotypes for Enhanced Growth and Yield Attributes Efficiency Index in Irrigated Lowlands of the Indo-Gangetic Plains.
2. Do you have any data about the analysis of leftover N in field, after crop harvest, that would give clear picture of NUE, otherwise the data mentioned is sufficient.
3. Please mention latest references (if available) 2020 onwards, you have mentioned some old references.
4. Line 86, pl. explain the purpose of neem oil-coated urea, for the benefit of the audience.
Reviewer 3 Report
1. Is the Title appropriate to describe whole story of the research? Can be Improved. In the title is using too much “yield” term.
2. Does the Abstract represent the research? Can be improved. Please add the Background information why did you choose Indo-Gangetic Plains and related to the Nitrogen fertilizer application. It is better if you add brief information related to the total number of rice genotypes that you used for the research. And also please add the statistical design for the field-based evaluation.
3. Does the Introduction explain the background of the research? Can be improved. It will be better if you add more background information from the previous studies related N use efficiency and associated that with the grain yield components.
4. Is the Objectives of the research address correctly? Yes.
5. What is the main question addressed by the research? Yes.
6. Is the main question relevant and interesting? Yes, it is relevant and interesting but need to improve the information in the introduction section.
7. How original is the topic? The topic is standard.
8. What does it add to the subject area compared with other published material? The subject is common because I found so many published articles with the similar topic.
9. Does the Methodology describe well? Can be improved. It is better if you put the Table 1 and 2 into Supplementary data.
10. Is the Result display in the correct way? Can be improved. It will be better if you also display the results in Diagram not only in the Tables.
11. Does the Discussion describe all of the results? Yes.
12. Is the paper well written? Yes, need improvement.
13. Is the text clear and easy to read? Can be improved. It will be better if the Text in the Tables use the Smaller Font Size.
14. Are the conclusions consistent with the evidence and arguments presented? Can be improved.
15. Are the conclusions address the main question posed? Can be improved.
16. Does the Reference cite appropriately? Yes.
Reviewer 4 Report
The manuscript entitled “Field-based evaluation of rice genotypes for enhanced growth, yield attributes, yield, and grain yield efficiency index in irrigated lowlands of Indo-Gangetic plains” The manuscript is full of technical and grammatical mistakes that should be improved before final publication in this journal. For improvement of manuscript, consider the following suggestions and comments.
Introduction: English language standard of this section is very poor and requires considerable improvement.
Many of the sentences are too long, having ambiguous language/words which makes it difficult for the reader to understand. Although this section provides a considerable detailed background of nitrogen use efficiency in rice and its variation due to genetic make up of the cultivars, however at line 90, authors stated that they will study low, half and recommended doses of N on various rice cultivars. While in materials and methods section, low N treatment was termed as “Control”.
Materials and Methods: Give more details about the agronomic practices applied and already recommended doses of Nitrogen to the rice crop in the area where experiment was conducted.
Results: Results of “ANOVA” are not given in the manuscript. Further, Pearson correlation was conducted by computing an average of 2 years data, while in previous sections, authors demonstrated that cultivars showed a significant differences for two years data for all the traits studied. How a significantly different datasets from 2 years can be averaged? Please provide detailed reasoning and logic to this methodology.
Discussion: This is the weak section of the paper, poorly written, and supported by very few references. More and recent citations needs to be added, and the language of this section (specifically) needs to be improved. At many points, authors used lengthy sentences, which make it difficult for the reader(s) to understands. Also noted, repetition of arguments without any logical justification.
Reviewer 5 Report
Dear authors,
I have studied your manuscript and it is well written and designed but it lacks novelty. The outcome of present study is known data and nothing new is presented in this manuscript, so I believe it does not meet the average to be accepted for publication in Sustainability,
I suggest publication of this manuscript in a local journal,
Good luck,
Reviewer 6 Report
The authors propose a manuscript titled “Field-based Evaluation of Rice Genotypes for Enhanced Growth, Yield Attributes, Yield, and Grain Yield Efficiency Index in Irrigated Lowlands of the Indo-Gangetic Plains”.
I suggest the following changes:
Introduction
The authors should provide more recent literature references.
Results
Please provide ANOVA tables with the mean squares and the significance with ** or * for 99% and 95% significance, respectively
Authors should also consider performing a combined analysis (years 2020 and 2021) and a multivariate analysis, in order to assess the possibility of interaction between genotype and environment.
Finally, discussion and conclusions should be rewritten in order to include the results of combined analysis.
Round 2
Reviewer 4 Report
The manuscript has been substantially improved after revision.
Author Response
*English revised throughout the paper. Discussion sections are improved fully.
Conclusions are as per line of the results now, and consistent.
All the suggestions incorporated.

Reviewer 6 Report
Introduction
The authors should provide more recent literature references.
Results
Authors should also consider performing a combined analysis (years 2020 and 2021) and a multivariate analysis, in order to assess the possibility of interaction between genotype and environment.
Finally, discussion and conclusions should be rewritten in order to include the results of combined analysis.
Author Response
Introduction
The authors should provide more recent literature references.
*Many references have been replaced [3,13 and 14] and new ones added [49 to 54 added] .
Results
Authors should also consider performing a combined analysis (years 2020 and 2021) and a multivariate analysis, in order to assess the possibility of interaction between genotype and environment.
*Above is complied. Kindly see the Table number 7 and 8 [Lines 296 – 299].
The results now described as per the new statistical analysis [ kindly see the lines 275 to 281]
Finally, discussion and conclusions should be rewritten in order to include the results of combined analysis.
*Discussion – Rewritten- kindly see the revised lines 339-342. Plus, one new heading (4.3 Interaction of years, nitrogen and genotypes) added to discuss the results [Lines 470-484].
Conclusion – revised fully. Kindly see the lines 502-509.

Round 3
Reviewer 6 Report
It was improved from the first version of the manuscript, however, references in the introduction section are not enough and adequate in order to predispose the reader of the article's theme. More relevant references need to be added.
Author Response
Nine references have been added in the introduction now. These are latest references and used appropriately at respective places. The article was read again and certain minor corrections made.
Following references added:
- Houlton, B.Z.; Almaraz, M.; Aneja, V.; Austin, A.T.; Bai, E.; Cassman, K.G.; Compton, J.E.; Davidson, E.A.; Erisman, J.W.; Galloway, J.N. A world of cobenefits: Solving the global nitrogen challenge. Earth’s Future 2019, 7, 865–872.
- Xu, P.; Chen, A.; Houlton, B.Z.; Zeng, Z.; Wei, S.; Zhao, C.; Lu, H.; Liao, Y.; Zheng, Z.; Luan, S. Spatial Variation of Reactive Nitrogen Emissions from China’s Croplands Codetermined by Regional Urbanization and Its Feedback to Global Climate Change. Geophys. Res. Lett. 2020, 47, e2019GL086551.
- Anas, M., Liao, F., Verma, K.K. et al. Fate of nitrogen in agriculture and environment: agronomic, eco-physiological and molecular approaches to improve nitrogen use efficiency. Biol Res 53, 47 (2020). https://doi.org/10.1186/s40659-020-00312-4.
- Sharma, L.K.; Bali, S.K. A Review of Methods to Improve Nitrogen Use Efficiency in Agriculture. Sustainability 2018, 10, 51. https://doi.org/10.3390/su10010051
- Wim de Vries, Impacts of nitrogen emissions on ecosystems and human health: A mini review, Current Opinion in Environmental Science & Health, Volume 21, 2021, 100249, ISSN 2468-5844,https://doi.org/10.1016/j.coesh.2021.100249.
- CGIAR-Consultative Group on International Agricultural Research, CGIAR Secretariat; Washington, D.C: 1998. CGIAR Annual Report.
- Debabrata Panda, Jijnasa Barik, Flooding Tolerance in Rice: Focus on Mechanisms and Approaches, Rice Science, Volume 28, Issue 1, 2021, Pages 43-57, ISSN 1672-6308, https://doi.org/10.1016/j.rsci.2020.11.006.
- Singh, B., Mishra, S., Bisht, D.S., Joshi, R. (2021). Growing Rice with Less Water: Improving Productivity by Decreasing Water Demand. In: Ali, J., Wani, S.H. (eds) Rice Improvement. Springer, Cham. https://doi.org/10.1007/978-3-030-66530-2_5.
- Neeraja, C.N., Voleti, S.R., Subrahmanyam, D., Surekha, K. and Rao Indian, P.R. Breeding rice for nitrogen use efficiency. Indian J. Genet., 79(1) Suppl. 208-215 (2019) DOI: 10.31742/IJGPB.79S.1.11
